# Human-Aware Control for Physically Interacting Robots

**DOI:** 10.3390/bioengineering12020107

**Published:** 2025-01-23

**Authors:** Reza Sharif Razavian

**Affiliations:** Mechanical Engineering Department, Northern Arizona University, Flagstaff, AZ 86011, USA; razavian.reza@nau.edu

**Keywords:** physical human–robot interaction, nonlinear model predictive control, optimal control, predictive modeling, human motor control, musculoskeletal model of movements, neuroscientific model of movements

## Abstract

This paper presents a novel model for predicting human movements and introduces a new control method for human–robot interaction based on this model. The developed predictive model of human movement is a holistic model that is based on well-supported neuroscientific and biomechanical theories of human motor control; it includes multiple levels of the human sensorimotor system hierarchy, including high-level decision-making based on internal models, muscle synergies, and physiological muscle mechanics. Therefore, this holistic model can predict arm kinematics and neuromuscular activities in a computationally efficient way. The computational efficiency of the model also makes it suitable for repetitive predictive simulations within a robot’s control algorithm to predict the user’s behavior in human–robot interactions. Therefore, based on this model and the nonlinear model predictive control framework, a human-aware control algorithm is implemented, which internally runs simulations to predict the user’s interactive movement patterns in the future. Consequently, it can optimize the robot’s motor torques to minimize an index, such as the user’s neuromuscular effort. Simulation results of the holistic model and its utilization in the human-aware control of a two-link robot arm are presented. The holistic model is shown to replicate salient features of human movements. The human-aware controller’s ability to predict and minimize the user’s neuromuscular effort in a collaborative task is also demonstrated in simulations.

## 1. Introduction

As robots become more integrated into human environments, ensuring that they can interact with people safely and intuitively becomes increasingly important. This is a challenging control problem that is especially critical in physical human–robot interactions.

In control systems, high-quality performance is only possible if the dynamics of the system being controlled is adequately modeled. This also applies to the control of human–robot interactions—to control coupled human–robot dynamics, it is essential to model the entire interaction, including the user’s movements. For example, in wearable robots, exoskeletons, surgical robots, and industrial collaborative robots (cobots), where the human user and the robot share a physical task, the robot must be able to predict the user’s behavior and act in a predictive rather than reactive way. To achieve this predictive control, mathematical models must be developed to predict human movements based on minimal information, such as the task’s high-level goals. Additionally, these models must be able to run many times within a single cycle of the robot’s controller to allow the robot to compare predictions and choose the best course of action.

Researchers in the fields of biomechanics and motor neuroscience have sought to present such predictive models for human movements, and significant achievements have been made. In musculoskeletal biomechanics, and especially gait biomechanics, large-scale nonlinear musculoskeletal models that rely on nonlinear optimization algorithms have demonstrated great success in predicting movement kinematics [1], muscle activity patterns [2], joint contact forces [3], and human–machine interaction [4] (see the review papers [5,6,7] for more success stories). These biomechanical models are rich in details, such as musculoskeletal anatomy and the mechanical properties of the limbs; however, the downside of these models is that they are slow to run—it may take minutes to hours to predict a few seconds of movement. Therefore, the computational efficiency of these models is a limiting factor that prevents them from being used for real-time predictive control applications.

On the other end of the model complexity spectrum, neuroscientific models of movement control seek to trim as much detail as possible and present a minimalistic model that captures salient features of human motor control. A prominent example of such models is based on stochastic linear optimal feedback control theory [8,9], which provides a mechanistic explanation of how an internal model of the world is used to process sensory information and plan and execute movements. Despite their simplicity, these neuroscientific models are invaluable since they explain why we move the way we do. Therefore, these models succeed at an objective that is different from that of biomechanical models. The former discusses the “why”, while the latter focuses on the “how”.

The challenge is that human movements are the emerging behavior of a complex dynamical system that constitutes the neural, muscular, skeletal, and sensory systems, which interact with the environment in a closed-loop manner. To have a truly predictive model, we need to take into account both the high-level (the why) and low-level (the how) details in a holistic model.

In this paper, I present a novel model for human movement with these considerations in mind. This holistic model encompasses multiple levels of detail in the human sensorimotor system; it is based on neuroscientific theories such as internal models and optimal sensorimotor integration for decision-making and action planning, as well as biomechanically plausible details of the musculoskeletal system. Therefore, this model enables predicting and distinguishing the contributions of various levels of the sensorimotor system to the user’s behavior. Importantly, because this model employs biologically plausible control processes, as opposed to nonlinear optimization in existing musculoskeletal simulations, it is computationally efficient and implementable in real time. At the same time, the model is also more accurate in its predictions, e.g., when dealing with new environments, because of its reliance on well-established neuroscientific theories.

In addition to the model, I also present the preliminary results of a new control paradigm for human–robot interaction, called human-aware control. This controller, which is based on nonlinear model predictive control, employs the developed holistic model of human movement to predict the user’s behavior during the interactions and optimize the robot’s joint torques in real time. This control paradigm is advantageous over the existing impedance/admittance-based controllers, e.g., [10,11,12], since the robot no longer treats the interaction as a disturbance to be observed and compensated. This human-aware controller is also advantageous over other predictive controllers that use data-driven models, e.g., [13,14,15], because it is not bound to the scope of the training data—the developed holistic model, being mechanistic as opposed to data-driven, can reliably predict movements in new interaction dynamics. In this paper, I present simulations of the human-aware controller for a two-link robotic arm as a proof of concept. To my knowledge, this human-aware control paradigm is the first control method that brings biomechanical- and neuroscientific-backed theories to the predictive control of physical human–robot interactions.

## 2. Methods

The human-aware control paradigm (Figure 1) consists of two major components: (1) a holistic model for human movement, which allows the robot controller to predict how the user reacts to the robot’s actions, and (2) an optimal controller, based on nonlinear model predictive control, which employs the holistic model to predict the future behavior of the user and optimize motor torques.

### 2.1. Holistic Model of Human Movements

Despite the nonlinearities and redundancies present in the musculoskeletal system, the nervous system manages to control human movements dexterously and in real time. Therefore, replicating the same control principles in silico is the best strategy to predict human movements in a computationally efficient and biologically plausible way. To this end, a holistic model for human movements is developed (Figure 2), which is based on existing theories in motor neuroscience and biomechanics.

This holistic model has a hierarchical structure representing different “levels” of control in the nervous system, which work together cohesively in a closed-loop manner (Figure 2B). The highest level of control (described in Section 2.1.1) is an abstract computational module that deals with action planning and decision-making. These abstract decisions are made based on a low-dimensional internal model [16] that resides in a simplified and abstract task space. This module receives sensory feedback from the body and optimally infers the current states of its internal model. Using these estimated internal states, an optimally tuned feedback control gain calculates some abstracted muscle commands in the low-dimensional task space. This low-dimensional output is then expanded into high-dimensional physiological muscle space in a mid-level module (Section 2.1.2) by utilizing coordinated muscle activity patterns, often known as muscle synergies [17]. Finally, the low-level biomechanical model (Section 2.1.3), which represents the nonlinear dynamics of the muscles, arm, and environment, receives muscle activities and produces movements in the physical world. The sensory information about the kinematics of the hand (the physical task space) is fed back to the higher-level modules to close the loop.

In the case study presented in this article, movements in 2-dimensional space, e.g., over a tabletop, are considered; however, extensions to movements in 3 dimensions and with kinematic redundancies are possible without modifying the framework [18,19,20].

#### 2.1.1. The High-Level Module: Decision-Making Based on Internal Models

This module is based on mounting evidence in motor neuroscience research, which supports the argument that humans chose a movement strategy that minimizes a combination of effort and task accuracy [8,9,21]. These models often focus only on the “task-level” dynamics—they abstract the entire movement, such as reaching, into a point-mass being moved around in a 2- or 3-dimensional task space. Although these highly simplified models fall short of representing the entire complexity of the neuromuscular system, their success in replicating patterns of movement in the task space is a strong indicator of their validity as a fundamental mechanism for movement-related decision-making [22,23,24]. In addition to their validity, the computational simplicity of these models makes them an ideal choice for the high-level decision-making module in the holistic model.

This high-level module is an extension of linear quadratic Gaussian (LQG) control, which is adapted to the multiplicative characteristics of the noise in the sensorimotor system—the standard deviation of noise scales linearly with the amplitude of the signal [25,26]. In this high-level controller, the internal model, i.e., the model based on which motor decisions are made, is an abstract 2-dimensional system consisting of a point-mass actuated by two “abstract muscles” (Figure 3). This abstract task space represents the most fundamental dynamics of movements in 2-dimensional space: Newtonian mechanics (i.e., inertial effects) and neuromuscular mechanics (i.e., muscle force production dynamics). Note that these abstract muscles can push and pull, unlike physiological muscles that are pull-only actuators. The equations of motion for this internal model along the *x* and *y* dimensions are as follows.(1)$$m\ddot{x}=a_{x}F_{max},\;m\ddot{y}=a_{y}F_{max}$$
where $F_{max}$ is the maximum muscle force that is scaled by the muscle activations, *a*, which follow first-order dynamics:(2)$$\tau \dot{a_{x}}=u_{x}-a_{x},\;\tau \dot{a_{y}}=u_{y}-a_{y}$$
Here, τ is the muscles’ activation time constant, and *u*s are the abstract muscle activation commands that the high-level controller determines. The numerical values of all parameters are provided in Table 1.

To solve the high-level control problem, this internal model is transformed into a discrete state-space form:(3)$$X_{k+1}=AX_{k}+\mathrm{Bu}+BCu+\xi$$
with $X={\left[x\;y\;\dot{x}\;\dot{y}\;a_{x}\;a_{y}\right]}^{T}$ and $u={\left[u_{x}\;u_{y}\right]}^{T}$ being the state and input vectors, respectively, and A,B being the corresponding discrete state and input matrices. The subscript *k* represents the $k^{th}$ time step. In (Equation 3), it is assumed that the dynamics of this internal model is subject to multiplicative noise (through the noise term C in BCu) and additive noise (ξ), both with Gaussian distributions. These noise characteristics in the internal model are essential as they are shown to be important contributors to human behavior [25,27].

The high-level controller calculates the optimal muscle commands to drive the internal model to a target state. The optimal commands are calculated by minimizing the quadratic cost function:(4)$$J=\sum_{k=1}^{N}X_{k}^{T}Q_{k}X_{k}+u_{k}^{T}Ru_{k}$$

Here, $Q_{k}$ and R are the state and control penalty weights. The control penalty R is constant, but the state penalty $Q_{k}$ is time-dependent; it is zero throughout the motion and only contains non-zero elements along its main diagonal in the last *D* time steps (dwell time) to penalize the position and velocity at the target. In other words, the high-level controller seeks to reach the target within the final N−D time steps with minimal effort, but the trajectory is unspecified.

Given the state dynamics (Equation 3) and quadratic cost function (Equation 4), the optimal high-level control law becomes(5)$$u_{k}^{*}=L_{k}^{*}{\hat{X}}_{k}$$
with $L^{*}$ being the optimally tuned feedback gain (see [26] for derivation). $\hat{X}$ is the estimated state vector, which is inferred from the delayed and noisy sensory information Y using a Kalman filter.(6)$${\hat{X}}_{k+1}=A{\hat{X}}_{k}+Bu_{k}+K_{k}\left(Y_{k}-H{\hat{X}}_{k}\right)+\eta$$
where K is the Kalman filter gain [26]. This state estimation process is also assumed to be contaminated with additive Gaussian noise, η. To calculate the Kalman gain, the sensory information Y is assumed to be a full readout ($H=I_{6\times 6}$) of the internal state vector that is contaminated with noise and delay:(7)$$Y_{k}=HX_{k-d}+\omega$$
with *d* representing the sensory delay and ω being Gaussian noise. However, during movements, the sensory information from the musculoskeletal model is used (see Section 2.1.3). Note that these sensory feedbacks correspond to biological sensory organs, including vision (hand position and velocity), muscle spindles (muscle length, shortening velocity, and rate of force [28]), and Golgi tendon organs (muscle force [29]).

Because of the multiplicative noise in the state-space equations (Equation 3), the optimal controller and Kalman gains must be obtained iteratively [26].

The estimated internal states in the abstract task space, $\hat{X}$, are considered the perceived motion, i.e., this is how the high-level controller (the “brain”) thinks the movement is progressing. Based on these perceived states of the world, the high-level controller calculates the abstract muscle commands $u^{*}$ to move the internal model toward the intended target. These low-dimensional muscle commands in the abstract task space must be translated into physiological muscle excitation commands e. This is performed in the mid-level module.

**Table 1 bioengineering-12-00107-t001:** Numerical values of parameters in the simulations.

**Internal model**			
Parameter	Value	Parameter	Value
*m*	3 kg	ξ	${\mathrm{diag}}_{6\times 6}$($2\times 10^{-7}$)
τ	30 ms	ω	${\mathrm{diag}}_{6\times 6}$($10^{-6}$)
$F_{max}$	1000 N	C	${\mathrm{diag}}_{2\times 2}$($2\times 10^{-1}$)
R	${\mathrm{diag}}_{2\times 2}^{*}$(1)	η	${\mathrm{diag}}_{2\times 2}$($10^{-9}$)
$Q_{k<N-D}$	${\mathrm{diag}}_{6\times 6}$(0)	*d*	50 ms
$Q_{k\geq N-D}$	${\mathrm{diag}}_{6\times 6}$(1)		
**Musculoskeletal model**			
Model as described in Section 4.1.6 in [30]			
**Robot model**			
Parameter	Value	Description	
$l_{1}=l_{2}$	0.3 m	Length of each link	
$m_{1}=m_{2}$	0.5 kg	Mass of each link	
$d_{1}=d_{2}$	0.15 m	Distance to center of mass	
$I_{1}=I_{2}$	$5\times 10^{-3}$ kg.m^2^	Moment of inertia about center of mass	
**Human-aware robot controller**			
Parameter	Value	Description	
Δt	5 ms	Discretization step size	
$N_{p}$	50 time steps	Prediction horizon length	
*D*	21 time steps	Dwell time at target	
*N*	221 time steps	Total duration of simulation	
**Impedance robot controller**			
Parameter	Value	Description	
$k_{p}$	50 N/m	Effective end-effector stiffness	
$k_{d}$	10 N.s/m	Effective end-effector damping	

^*^ A diagonal matrix of a given size with the parameters on the diagonal.

#### 2.1.2. The Mid-Level Module: Dimension Expansion with Muscle Synergies

The high-level controller’s abstract and low-dimensional commands must be expanded into the high-dimensional muscle space to be physiologically realistic. The first challenge is that physiological muscles can only pull, unlike abstract muscles that can pull and push. Further, the human musculoskeletal arm has more muscles than degrees of freedom, leading to infinite possible solutions for muscle forces that can produce a specific task space force. Lastly, the nonlinear dynamics of the multi-link arm, as opposed to the linear dynamics of the point mass, must be taken into account to produce accurate movements. Therefore, solving for muscle forces that replicate the task-space movement as predicted by the internal model is not straightforward. This problem—referred to as the muscle load-sharing problem in biomechanics—is usually solved using nonlinear optimization [5,6], a process that is neither physiologically plausible nor implementable in real time. Instead, this holistic model employs a biologically plausible approach to solving muscle activities using the well-supported theory of muscle synergies [17].

An overview of the dimension expansion module is shown in Figure 4. In this approach, muscle synergy is defined as the co-activation of the *m* muscles in the musculoskeletal system with known relative ratios, represented by the vector $S_{m\times 1}$ [31] (Figure 4A; also see Appendix A for the calculation method). It is postulated that the nervous system holds a memory of synergies and their actions in the task space, i.e., the nervous system knows the hand force vectors produced by each synergy (Figure 4B). These synergy-produced force vectors, $B_{i}$, form a basis set for the task space [19,20]. The abstract muscle command in the low-dimensional task space is projected onto these basis vectors to calculate the coefficient of the force along each basis. Because there are usually more synergies (4 in the presented example) than dimensions of the task space (n=2), the decomposition solution is not unique; therefore, a non-negative least-squares algorithm is used to solve for the non-negative coefficient vector $C={\left[C_{1},\ldots ,C_{4}\right]}^{T}$ in the following equation:(8)BC=u

Here, $B_{2\times 4}=\left[B_{1}\left|\ldots \right|B_{4}\right]$ is the matrix formed by the 4 synergy-produced basis vectors, $B_{i}$, in the 2-dimensional task space, and $u_{2\times 1}$ is the abstract muscle command from the high-level controller. Once the coefficients $C_{i}$ are calculated, the weighted sum of the synergies (Figure 4C) results in the “neural” excitation commands $e_{6\times 1}$ that represent aggregate motoneuron activities.(9)e=SC

Here, $S_{m\times 4}=\left[S_{1}\left|\ldots \right|S_{4}\right]$ is the synergy matrix containing all synergy vectors.

The calculated neural excitation commands can be directly used as the input to the musculoskeletal model (see Section 2.1.3) to approximately produce the task-space forces directed by the high-level controller. However, because of the nonlinear dynamics of the multi-link arm, velocity-dependent accelerations lead to a task-space motion that is different from the desired one. To overcome that, the high-level muscle commands are adjusted using the known dynamics of the arm to compensate for the velocity-dependent acceleration. For details, readers are referred to [32].

An offline optimization and data reduction procedure can be used to obtain the synergies. The details are available in [19,32] and are omitted here for brevity. This offline approach to learning and storing synergies parallels skill acquisition and motor memory formation in humans [33].

#### 2.1.3. The Low-Level Module: Musculoskeletal and Environment Dynamics

The mid-level module’s output is a set of neural excitation commands, e, that are fed to the musculoskeletal model. In this case study, the musculoskeletal model representing the body is a 2-link planar linkage that is actuated by 6 muscles (Figure 5). This model type is the standard upper extremity model for various biomechanical and motor control studies [18,34,35,36], as it captures the important features of a human musculoskeletal system: muscle redundancy, nonlinear muscle mechanics, and the nonlinear dynamics of the arm. The two links in the model represent the upper arm and the forearm, which are connected to a fixed torso and to each other via revolute joints. The wrist is not modeled. The muscles are modeled as nonlinear Hill-type muscles [37] that produce the force *F* according to(10)$$F=af_{l}f_{v}F_{max}$$
with $f_{l}$ and $f_{v}$ being the force–length and force–velocity relationship [37]. a∈[0,1] is the activation of the muscle, and $F_{max}$ is maximum isometric muscle force. The muscle activation is driven by the neural excitation signal *e* according to the dynamics(11)$$\dot{a}=\left\{\begin{array}{cc} \frac{e-a}{\tau_{act}\left(0.5+1.5a\right)} & e>a\\ \;\\ \frac{e-a(0.5+1.5a)}{\tau_{deact}} & e\leq a \end{array}\right.$$
with $\tau_{act}$ and $\tau_{deact}$ being the muscles’ activation and deactivation time constants. The contractile forces (Equation 10) are applied to arm links along each muscle’s line of action, which is defined as the straight line connecting the origin and insertion points of the muscle. The tendons and other compliances of the muscle are neglected in this musculoskeletal model. The musculoskeletal model parameters are taken from [30].

The sensory information from the musculoskeletal arm closes the control loop in the holistic model (Figure 2B). The high-level controller receives the delayed feedback (with the same delay time, *d*, as in (Equation 7)) from the musculoskeletal hand position and velocity to estimate the internal model’s states according to (Equation 6). Therefore, the estimated internal states are updated based on the actual movement’s measurements rather than the internal model’s state dynamics.

The holistic model of human movement can predict how movements evolve, only using a high-level objective of the task, e.g., move to location X. Importantly, because of its feedback structure, the model responds to interactions with the environment in a human-like way (see the Results).

To create an “environment” with which the holistic model can interact, the “hand” of the musculoskeletal arm is attached to the end-effector of a 2-link planar robotic arm with a revolute joint (Figure 5). Different interactive scenarios can be simulated by controlling the robot (see Section 3). One such controller is the developed human-aware controller (Section 2.2), which determines the robot joint torques, *T*, to alter the human’s behavior in a targeted way.

### 2.2. Human-Aware Control of Robot

The human-aware control paradigm is based on a nonlinear model predictive control (NMPC) scheme [36,38] (Figure 1). The human-aware controller relies on the presented holistic model to internally run predictive simulations and forecast the user’s interactive behavior for a short time window in the future, called the prediction horizon. In a given time step during the motion, the human-aware controller initializes this model using the current measurements from the plant and runs predictive simulations for the duration of the prediction horizon $N_{p}$. The predictive simulation informs the controller how the human will continue the motion when faced with the interactive forces supplied by the robot. The controller calculates the motor torques, $T_{1}$ and $T_{2}$, to minimize an objective computed over the prediction horizon. Then, the controller applies these optimal torques to the robot for one time step; the entire process repeats in the next time step with updated measurements from the system.

The objective function may be a function of any variables in the holistic model, e.g., those in the high-, mid-, or low-level modules. This is an important feature of the human-aware controller, as variables that are not easily measurable (such as muscle forces or the internal model’s states) can be optimized. As a case study, a simple objective function considered in this study is the minimization of the user’s “effort”:(12)$$J=\frac{1}{N_{p}}\sum_{k=1}^{N_{p}}{\left|e_{k}\right|}^{2}$$
where $e_{k}$ is the vector of neural excitations in the holistic model. The summation is over the time steps *k* from the beginning to the end of the prediction horizon with the length $N_{p}$. To have smooth torque profiles over the prediction horizon, the robot’s motor torques are parameterized using the 2nd-order polynomial $T=\sum_{i=0}^{2}\beta_{i}k^{i}$, and the decision variables in this receding horizon optimization are the 6 coefficients $\beta_{i}$ for the two robot torques. The controller uses a nonlinear optimization algorithm to find the robot’s joint torque coefficients such that the objective function (Equation 12) is minimized. It must be noted that no specific condition is enforced on the torque profiles here, such as continuity or the rate of change. These additional considerations can easily be incorporated into the NMPC framework, either as hard constraints or soft constraints, i.e., including penalty terms in the objective function (Equation 12).

In making predictions in the prediction horizon, it is assumed that the robot is aware of the goal of the task: to reach a known target in the task space (Figure 5). Further, in the present implementation of the human-aware controller, it is assumed that the controller has access to the current measurement of the robot’s joint positions and velocities, as well as the human’s joint angle and velocities, muscle activities, and internal model’s states. It is not realistic to assume that the robot has a direct measurement of the human’s internal states, but in the future, nonlinear estate estimation techniques such as a moving horizon estimator [39,40] may be used to provide the controller with information about the states of the model as well as the user’s intent (which is encoded within the states of the internal model).

## 3. Simulation Settings

The holistic model of movements and the human-aware control paradigm were implemented and evaluated in simulations.

In the first set of simulations, only the holistic model was considered; the goal was to evaluate how well the model captured the salient features of human movements. In these simulations, well-documented “center-out” movements were replicated with the human movement model. The holistic model was provided with eight target locations, equally spaced around a circle of a 12 cm radius (consistent with prior experiments, e.g., [41]), which the model had to reach within 1 s. The predicted hand paths were expected to be straight lines from the center position to the targets [42], with velocity profiles that were smooth, bell-shaped, and symmetrical [43]. The muscle activities were also expected to be distributed across different muscles but with minimal agonist–antagonist co-contractions [1].

In addition to these unconstrained movements, the model was also placed in a “curl force field” [44]; a velocity-dependent perturbing force was applied to the model’s hand perpendicular to the direction of movement:(13)$$F_{pert}=b\left[\begin{array}{cc} 0 & 1\\ -1 & 0 \end{array}\right]\left[\begin{array}{c} \dot{x}\\ \dot{y} \end{array}\right]$$
with b=3N.s/m for clockwise and b=−3N.s/m for counterclockwise force fields. The model predicted movement trajectories for reaching the same eight targets while subjected to either force field. The model was expected to produce movement trajectories that were curved in the direction of the force field [41,44].

To test the human-aware controller, four simulations were run. **A. No interaction**: The first simulation was an undisturbed movement to reach a target 25 cm to the left within a prescribed time of 1 s. This motion was the baseline for comparing other interactive scenarios. **B. Robot off**: In the second simulation, the human model moved to the same target while maintaining physical contact with the robot, but the robot torques were set to zero. In this case, the human carried the inertia of the robot (with its nonlinear effects). **C. Human-aware control**: in this condition, the robot was controlled with the human-aware controller to assist with the task and minimize the user’s effort (Equation 12). **D. Impedance control**: In the last simulation, the robot was controlled by a mainstream impedance controller [45] to compare its assistive properties to that of the human-aware controller. To create an “assistive” effect, the impedance controller was set to follow the reference trajectory obtained in the “no interaction” scenario **A**; however, the trajectory was advanced 50 ms in time to pull the hand toward the target and provide assistance. The impedance parameters were chosen to represent “human-like” stiffness and damping [24].

The input in simulations A–C was only the target position and movement duration. The fourth simulation with the impedance controller also required the reference trajectory as an input, which was obtained from the first simulation. All simulations were performed in MATLAB 2024a (Mathworks Inc., Natick, MA, USA), with the numerical parameters in Table 1.

## 4. Results

### 4.1. Predicting Movements with the Holistic Model

The first set of results focuses on the predictive abilities of the holistic model. In center-out motion simulation (Figure 6A), the model predicted straight hand paths from the center position to the targets, and the transitions were smooth with bell-shaped velocity profiles, consistent with experimental observations [42,43]. As shown in the two exemplary motion directions, the high-level controller was successful in correctly estimating the states of the motion (note the similarity of trajectories in the “abstract” and “physical” task spaces), despite the simplicity of the linear internal model. Further, the mid-level module could successfully expand the abstract muscle commands into high-dimensional neural excitations that produced muscle force with minimal agonist–antagonist co-contraction (note the alternative pattern of muscle activities in the acceleration and deceleration phases of movement). The center-out results further show that the holistic model could distribute the required load among muscles with small agonist–antagonist co-activations and move the arm toward the intended target.

When the hand motion was perturbed by the curl force field, the hand trajectories were deflected in the characteristic curved paths (Figure 6B,C), consistent with experimental observations [41,44]. According to these results, the muscle activities changed through the feedback control structure to move the hand toward the target despite the initial deflection. These results demonstrate that the holistic model can predict human-like behavior during unperturbed reaching movements and interactions with unknown environments. Note that the holistic model was only supplied with the goal of the task: move to the intended target. How to move there and deal with unknown perturbations was determined through the model’s multi-level structure.

### 4.2. Human-Aware Control Results

The second set of results (Figure 7) regard the human–robot interaction. Like according to the previous results, the unconstrained reaching movement to the target exhibited human-like features (Figure 7A). Interactions with an unactuated robot in the second simulation (Figure 7B) introduced a visible lag in movement (note the delayed peak velocity) and increased muscular effort to accelerate and decelerate the added inertia of the robot. Despite the “unknown” dynamics of the robot, the arm arrived at the target because of its feedback structure.

The introduction of the human-aware controller in the third simulation significantly altered movement patterns (Figure 7C). In this simulation, the human’s neuromuscular controller always sought to compensate for the unknown interactions with the robot. However, the robot’s human-aware controller could predict these compensations and optimally adjusted the interaction forces to minimize muscular effort, albeit at the cost of unnatural movement patterns. In this case, the optimal robot action involved rapid acceleration (much faster than the normal human acceleration in Figure 7A), followed by gradual deceleration. Consequently, the neuromuscular activities decreased significantly during both the acceleration and deceleration phases (Figure 7C). This effort-optimal behavior contrasts sharply with the movement patterns that emerged from the impedance control in the fourth simulation (Figure 7D). Despite the time shift in the reference trajectory, the human model quickly caught up with it and continued the movement with the robot. Therefore, the robot and the human model effectively moved their own inertias. The robot did not assist the user in this case, evident by the unchanged movement patterns and neuromuscular activities (compare panels A and D of Figure 7).

## 5. Discussion

In this paper, I presented a predictive model for human movement with two objectives: the ability to predict movements with biologically realistic neuromuscular attributes and the ability to make such predictions faster than real time for robot control purposes. I also presented a novel human-aware control paradigm, which employed the developed model for optimal control of human–robot interactions.

The holistic human movement model was truly predictive and only required the task description—in this case, the target position and movement time—to predict the kinematics and neuromuscular activities similar to experimental data [42,43]. The model could also handle the unmodeled dynamics and disturbances and finish the tasks in a human-like manner. Although the holistic model remains to be rigorously validated against detailed experimental data, individual modules within the model were based on well-supported biomechanical models and neuroscientific theories, contributing to the model’s validity as a whole. Specifically, the notions of internal models [16], optimal estate estimation [9], and optimal motor planning [9] are widely discussed in the computational neuroscience studied and are backed by experimental data. Further, the coordinated activity of muscles has been extensively documented [46,47,48], leading to the realistic reconstruction of muscle activities using a small set of synergies. Lastly, the inclusion of the physiological characteristics of the musculoskeletal system [6,37,49] contributes to the bio-fidelity of the entire model. This holistic model, however, still misses important physiological attributes that shape human movement, e.g., reflex structures [50] and mechanical stiffness [51], which can improve predictions of physical human–robot interaction, especially in response to sudden disruptions [24].

There often is a trade-off between accuracy and computational speed in mathematical models; however, this holistic model could closely replicate human behavior while maintaining computational efficiency. The holistic model’s accuracy and computational efficiency are due to its biologically plausible hierarchical structure—the nervous system controls movements in real time, and replicating them in silico is the best approach in improving both speed and accuracy. To give the results more context, it took 350 ms to simulate a 1.2 s movement time with 5 ms time steps (on a laptop with Core i7 7660U CPU and 16 GB of RAM, running MATLAB 2024a), making the model 3.4 times faster than real time. In comparison, state-of-the-art predictive musculoskeletal simulations are still 20–45 times slower than real time [36,52,53]. It must also be noted that, unlike existing musculoskeletal simulations in which computation time increases with the model’s state and control dimensions, the hierarchical feedback control structure in the holistic model is highly scalable. High-level computations only involve simple matrix multiplications in the control gain and the Kalman filter. Mid-level computations are also scalable, involving solving a system of linear equations using the non-negative least-squared method. Integrating the multibody musculoskeletal system with Hill-type muscle models is the most computationally costly process.

Despite the holistic model’s computational efficiency, the human-aware controller is still not implementable in real time. The nonlinear optimization in the model predictive controller takes approximately 1 s to converge to the optimal torques in each time step. Improvements to the model and the human-aware controller’s computational efficiency are still needed to make the robot controller operate in real time.

The holistic model and the human-aware controller included many parameter assumptions that could affect the results and need careful consideration. In the holistic model, all movement control decisions were made based on the internal model. This internal model represented the simplified movement dynamics; however, how simple or complex internal models are remains largely unknown in motor neuroscience. In this work and other studies, e.g., [22,23,27,54], three fundamental properties—Newtonian mechanics, muscle dynamics, and signal-dependent noise—seem sufficient to produce human-like movement trajectories. Similarly, in the mid-level dimension expansion module, the number of synergies is a critical parameter, which is also debated. The usual practice in the literature to tackle this question involves an inverse approach, in which synergies are determined through dimensionality reduction applied to recorded muscle activities; the smallest number of synergies that capture 90% of the variance of the data is taken [46,55]. However, this approach is purely data-driven without significant scientific rigor. In the holistic model, a forward approach was taken [19], in which the number of synergies that most closely reproduced the optimal muscle activities were chosen. In this work, physiologically realistic parameter values from the literature were used to show the working principles of the holistic model and the human-aware robot controller. Further investigations are needed to understand the effects of different parameters and assumptions on the results. Formal sensitive analysis can reveal the most critical parameters in the model, which may also produce testable hypotheses to uncover the mechanisms of the human sensorimotor control system.

A major assumption in the human-aware controller was that the robot had perfect knowledge of the collaborative task, i.e., the target’s location, as well as the movement’s onset and duration. An ideal human-aware controller must be able to infer such information from the interaction—the same way two people exchange information during a task, even without verbal or nonverbal communication and only through the sense of touch. Further, the human-aware controller was assumed to have access to the measurements from the user, e.g., its internal states, which are fundamentally unmeasurable. A possible workaround for both challenges involves using a state estimator, such as a moving-horizon estimator [39,40], to estimate the user’s states from the kinematics and interaction forces. Lastly, the human-aware controller assumed a “naive” user, i.e., the user was not expecting any interaction force during the movement. However, the human-aware controller may include human–robot co-adaptation. The interaction forces from the robot can be modeled as a state-dependent force, $F_{robot}=A_{k}X_{k}$, that may be learned by the human; its parameters can be included in the internal model (Equation 3) and iteratively updated using movement error [56]. Likewise, the model inside the human-aware controller can be updated iteratively as the user continues to react to the robot, e.g., using homotopy optimization [57].

The objective of the human-aware controller in this report was to minimize the user’s effort, which is only one of the many possibilities. Once the robot’s controller has access to a mathematical representation of individual aspects of the neuromuscular control system, e.g., the internal model’s states or the muscle synergies, the human-aware controller may seek to minimize any function of those variables; for instance, it may increase or suppress the activation of a muscle synergy for motor learning or rehabilitation.

## 6. Conclusions

This paper presented a predictive model for human movement and a novel human-aware control paradigm for human–robot collaboration. Rooted in biomechanical and neuroscientific principles, the holistic model emulated human motor behaviors and enabled the real-time prediction of neuromuscular activities. Leveraging this model, the human-aware control algorithm optimized robot actions to reduce user effort in interactive tasks, as validated in simulations with a robotic arm. We must note that this proof-of-concept development has limitations that prevent the controller from being implementable in robots; further development is needed. Nonetheless, these preliminary results reveal a promising approach to enhance human–robot collaboration, setting the stage for future research to refine and expand its real-world applications.

## Figures and Tables

**Figure 1 bioengineering-12-00107-f001:**
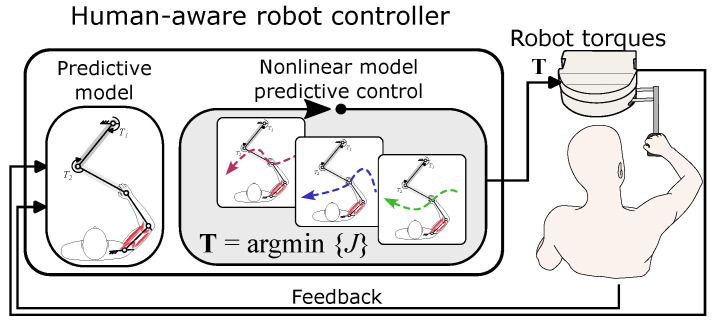
The human-aware control of a robot based on the nonlinear model predictive control scheme. A holistic model of human movement predicts the user’s behavior. The model is run iteratively to find the best robot actions.

**Figure 2 bioengineering-12-00107-f002:**
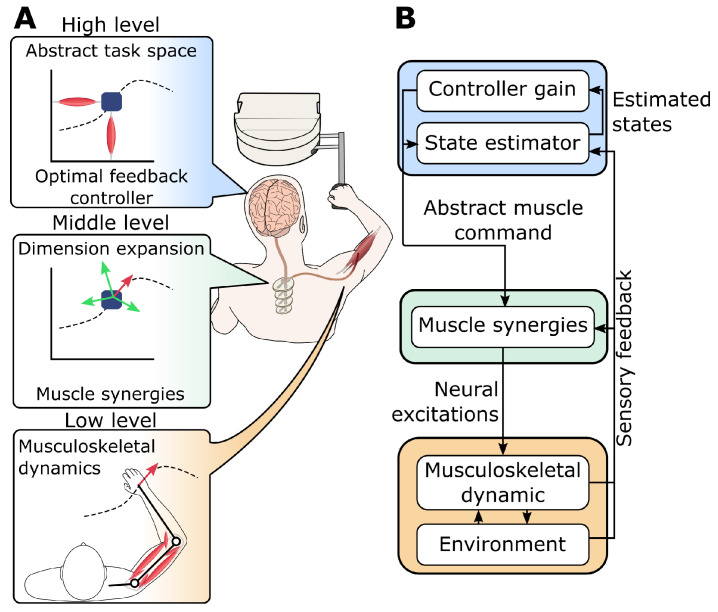
(**A**) An overview of the holistic model of the human movement. Each module represents a distinct level in the sensorimotor control hierarchy. (**B**) The hierarchical structure of the model based on biomechanical and neuroscientific theories and the information flow between modules.

**Figure 3 bioengineering-12-00107-f003:**
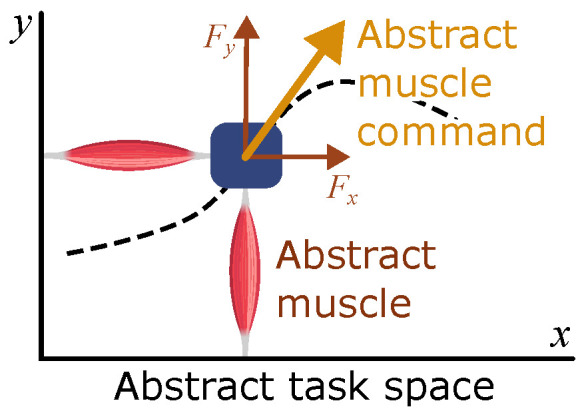
The internal model used in the high-level controller. The model is in the abstract 2-dimensional task space that represents the hand position. Two abstract muscles move a point mass in this space; the total muscle force is an abstract muscle command that is sent to the lower-level module.

**Figure 4 bioengineering-12-00107-f004:**
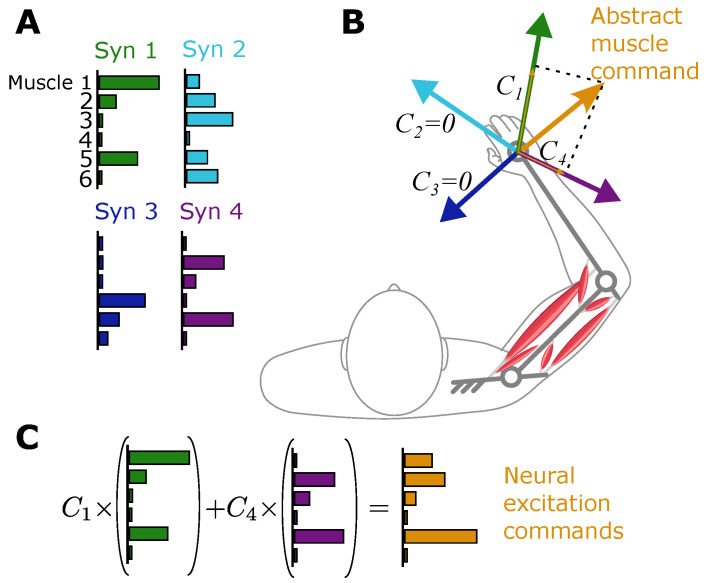
The mid-level dimension expansion module using muscle synergies. (**A**) Synergies are co-activations of muscles with known relative ratios. (**B**) The effect of each synergy in the task space is a force vector known to the motor controller. The low-dimensional neural excitation command in the task space is decomposed onto the synergy-produced bases to calculate the activation coefficient of each synergy, *C*. (**C**) Synergies are combined with the calculated coefficients to produce neural excitation commands.

**Figure 5 bioengineering-12-00107-f005:**
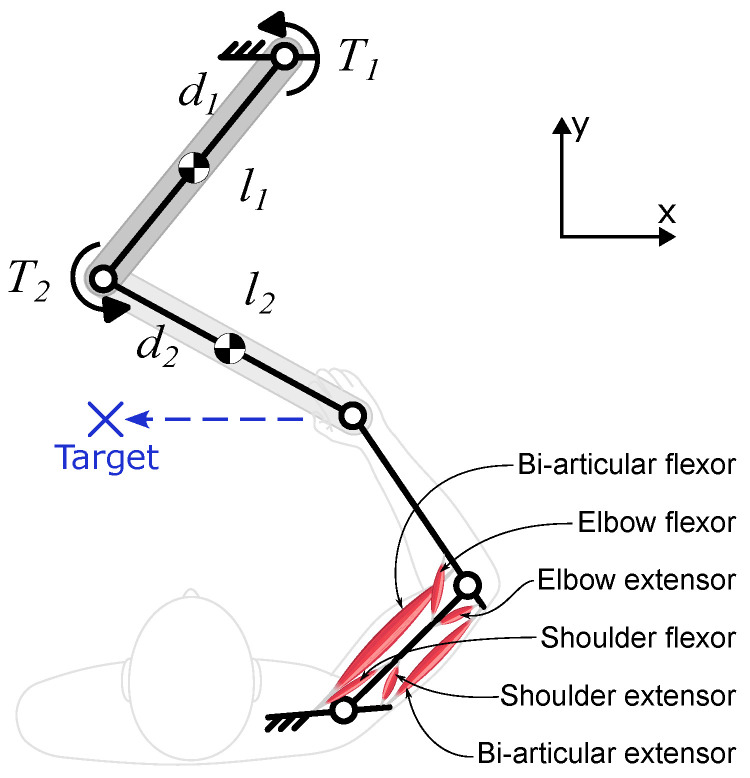
The musculoskeletal arm coupled with the 2-link robot.

**Figure 6 bioengineering-12-00107-f006:**
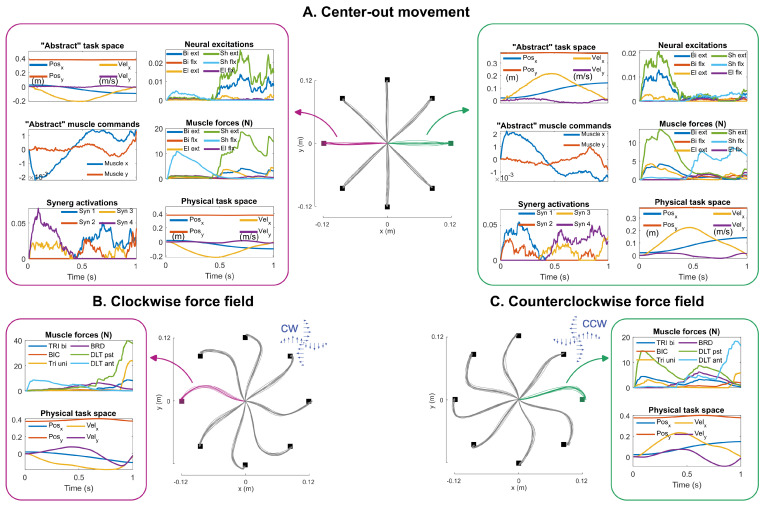
The simulation results of the holistic model of human movements. (**A**) The predicted “center-out” motion in the absence of any interactive force. The movement paths were straight with smooth velocity profiles; muscle activities showed minimal agonist–antagonist co-activation. The model predicted the “abstract” variables (related to the internal model) as well as the “physical” ones (related to the musculoskeletal system). Details of two exemplary directions are shown. (**B**) The movement trajectories during the clockwise force field. (**C**) The movement trajectories during the counterclockwise force field.

**Figure 7 bioengineering-12-00107-f007:**
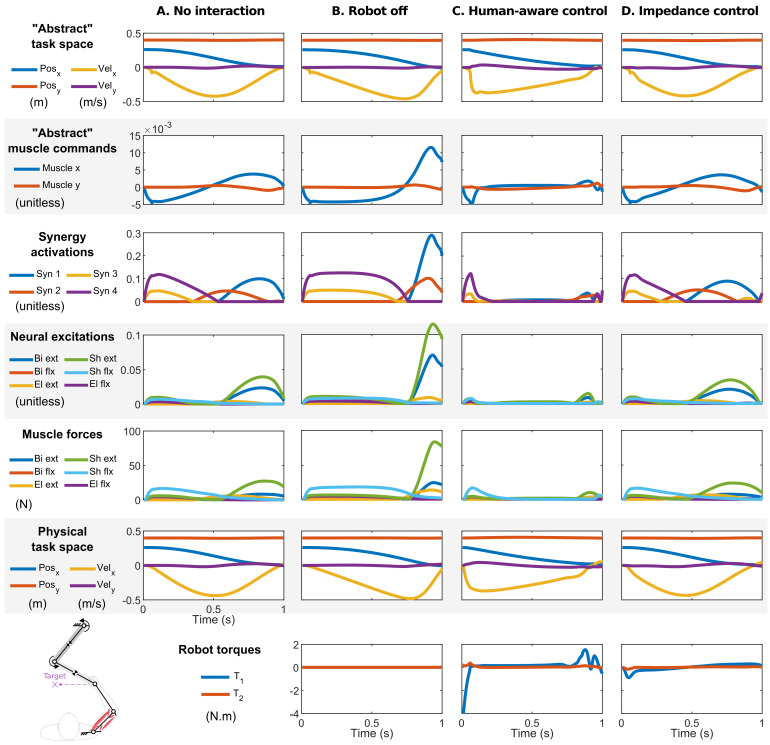
The simulation results of the human–robot interaction. (**A**) Human movement in isolation. (**B**) Human–robot interaction when the robot is turned off (zero motor torques). (**C**) Human–robot interaction with the human-aware controller. The robot’s objective was to minimize the human’s muscular effort. (**D**) Human–robot interaction with an impedance controller. The robot’s reference trajectory was the 50 ms time-shifted trajectory taken from the free-reaching movement (panel **A**).

## Data Availability

Codes for generating the results can be accessed from https://github.com/rsharifr/holisticMotorControl, accessed on 22 December 2024.

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
