# Peer review of "Human-Aware Control for Physically Interacting Robots"

_bioengineering, 2025, doi:10.3390/bioengineering12020107_

Round 1
Reviewer 1 Report
Comments and Suggestions for Authors
This article describes a novel model for predicting movements that is utilized for a control method with the application of human-robot interactions. I found the paper to be very well written and clear to understand with the figures exceptionally well done. The advantage of this model is in its high efficiency that is achieved by leveraging a hierarchical structure of decision making framework. The describes how this model can be used to guide a user's movement by minimizing their effort in the predicted path that is updated in based on a feedback mechanism. The major limitation of this work is the speed is still too slow to be used in real-time, which would allow it to have significant impact. However, given the wide spread potential applications of this work for safety, enhancement, and education/training, I still recommend this work for publication.
Author Response
Comment: This article describes a novel model for predicting movements that is utilized for a control method with the application of human-robot interactions. I found the paper to be very well written and clear to understand with the figures exceptionally well done. The advantage of this model is in its high efficiency that is achieved by leveraging a hierarchical structure of decision making framework. The describes how this model can be used to guide a user's movement by minimizing their effort in the predicted path that is updated in based on a feedback mechanism.
Response: Thank you for your time and feedback. I revised the manuscript to address your comments, and further responses are below.
Comment: The major limitation of this work is the speed is still too slow to be used in real-time, which would allow it to have significant impact. However, given the wide spread potential applications of this work for safety, enhancement, and education/training, I still recommend this work for publication.
Response: Thank you for the comment. You are correct that the current implementation of the method is indeed slower than real-time. Another major limitation, as I mentioned in the manuscript, is that the controller was assumed to have access to the “states” of the user, which is also not realistic (and I have discussed this issue in the Discussion section). More work is indeed needed to make the controller implementable in hardware.
However, it must be noted that the manuscript aims to present a new idea and a proof-of-concept implementation, not the complete implementation. The codes were written in MATLAB and were not optimized. Code optimization and reimplementation in faster languages (like C++) should make it significantly faster and closer to real-time. Model-based state estimation in combination with partial measurements from the interaction (e.g., interaction force and user’s motion) can also overcome the second limitation of access to the user’s states. These efforts will likely take a few more years of work.
I have revised the introduction and conclusion to highlight that this manuscript presents a proof-of-concept.
Reviewer 2 Report
Comments and Suggestions for Authors
Overall a well written report on the development and in silico testing of a 2D collaborative robot manipulator.
I think this paper does overstates the computational complexity of biomechanically based models in terms of computational time. The implementation of muscle based control for the collaborative tasks is indicative that these methods can be implemented in real time. Several of the references note real time applications of similar methods.
When reading the article I was expecting the task implementation to include higher dimensional control. A 2 Degree of freedom manipulator controlling a point mass on plane does not require high level control, as there is no redundancy in the system.
It also seems like you're saying that the robot controller knows the state space of the human partner actuator? How can this be outside of a simulation? Are you assuming a future visual system tracking the human partner an digitizing their pose in real time?
It seems like your 'human' model is just the biomechanical model / simulation. Which is a bit disappointing, it's hard to predict the utility of the control algorithm without real-world application.
Reviewer 3 Report
Comments and Suggestions for Authors
This paper proposes a modular human-aware controller for a simple human-robot interaction scenario. The paper is well-written and well-organized. The following points may be addressed to improve its presentation quality.
1. Suggest add sensitivity analysis with respect to sensory delay d.
2. Suggest use a different symbol than C to represent multiplicative noise in (3), as C is used for coefficient matrix later.
3. Suggest add subscript k to Y in (6).
4. It is not clear why a full readout in the state estimator (6) was assumed. In particular, how is hand acceleration sensed? Suggest choose a more physiologically relevent H matrix in (6).
5. Suggest provide the matrices used in (3)-(9) in an appendix. Explain how the basis matrix in (8) was obtained.
6. If beta_i are recomputed at each step (line 278), how is the continuity of T1 and T2 ensured?
7. How are human joint angles and velocities computed for use in the robot controller (line 286)?
8. I am wondering if you have tried the impedance controller in the presense of the curl force field.
9. How were feedback gains for the impedance controller calculated?
10. In Table 1, howcome l1,l2 are smaller than d1,d2?
11. What happens when dwell time is increased or decreased? The peaking behavior of shoulder externsor in Fig. 7B suggests that a longer dwell time for 'Robot off' is needed.
12. Correct typo on line 357.
13. Correct typo on line 396.
14. Correct typo on line 406.
15. How was best performance (line 433) assessed? Was it consistently 'best' across all tasks?
